# Intention to Use E-Payments from the Perspective of the Unified Theory of Acceptance and Use of Technology (UTAUT): Evidence from Yemen

**Fahd Alduais** [1,*] and **Mohammad O. Al-Smadi** [2]

1 Department of Accounting, Philadelphia University, Jarash Road, 20 KM, Amman 19392, Jordan
2 Department of Finance and Banking Sciences, Philadelphia University, Jarash Road, 20 KM, Amman 19392, Jordan
* Correspondence: falduais@philadelphia.edu.jo

**Abstract:** This study explores the challenges facing the current e-payment systems and investigates the main factors that support using the e-payment system. This study used a cross-sectional approach. An online survey was conducted on Yemeni consumers as part of the collection of data. The data from 486 questionnaires were analyzed using smartPLS4 and Jamovi software for structural model analysis and statistical analysis. According to the PLS-SEM results, the structural model shows that $R^2$ is 0.757, which explains the variances in behavioral intention via all the constructs. Statistically, the intention to use e-payment systems is significantly and positively influenced by performance and effort expectancies and social influence. In contrast, facilitation conditions are significantly and negatively correlated with behavioral intention. This is attributed to consumers' view of the infrastructure of Internet services, which does not contribute to the behavioral intention and acceptance of using electronic payment in Yemen. Contrary to expectations, age does not moderate the relationship between performance expectancy, social influence, and intention to use e-payment systems; hence, the related hypothesis was not supported. This study provides valuable suggestions for policymakers, designers, developers, and researchers, enabling them to better understand the critical aspects of using the electronic payment system. This study developed a model for predicting the likelihood of acceptance of electronic payments in a country that has not given adequate attention to this issue. An application and evaluation of the UTAUT model in Yemen are presented in this study.

**Keywords:** technology acceptance; facilitating conditions; intention; e-payment; Yemen

## 1. Introduction

Globally, the technological revolution rendered technology an integral part of the daily life of society, and technology in our daily lives has become increasingly prevalent. Various fields, including research, business, economics, education, and health, have been improved due to the development of information and communication technologies (Malaquias and Hwang 2019; Bankole and Bankole 2017; Alduais 2013). As a result of technological development, electronic payments have emerged as one of the manifestations of technological development. The use of electronic payments is on the rise throughout the world (León 2021; Qin et al. 2017). Its success depends on the accessibility of new technologies, the changing lifestyle choices of consumers, and many economic factors (Liébana-Cabanillas and Lara-Rubio 2017). E-payments have become an increasingly popular trend in a rapidly expanding market. Further research is required to comprehend the adoption process of these tools and monitor the impact of different financial solutions on consumer perceptions and daily lives (Luna et al. 2019).

Payments for e-commerce purchases can be made in various ways, both on- and offline. Most retail businesses accept online payments, for which customers may pay through the Internet directly using credit cards, electronic cash, and smart cards (Iman

2018). Offline payments are not possible through the Internet; instead, customers must use other payment options, such as bank transfers, post office payments, and payment on delivery. It can be concluded that consumers will decide whether to use payment services via e-commerce based on their perception regarding two factors, which are the recognition of the benefits and usability of the system. E-payment systems have been widely studied in developed countries due to technological advancements in payment systems. Few studies have examined the acceptance of these systems in developing countries, such as those in the Arab region (Al-Okaily et al. 2020; Al-Ajam and Nor 2015). Yemen still falls behind in terms of using the Internet and electronic payment systems compared with its neighbors. Accordingly, the current study fills the gap mentioned above and identifies important factors that affect the acceptance of e-payment systems in Yemen. Providing and using the electronic payment system has become the main challenge for many developing countries, including Yemen.

The electronic payment system has many great features that may be valuable. However, the successful use of an electronic payment system depends on understanding the adoption factors and the main challenges facing current electronic payment systems. There is a lack of agreement about the challenges and critical factors that constitute the successful use of an electronic payment system, especially in a country like Yemen; hence, an evident knowledge gap has been identified regarding the challenges and critical factors of using electronic payment. This study aims to explore the critical challenges facing the current electronic payment systems and investigate the main factors that support using the e-payment system. Using UTAUT 2 by researchers such as (Yaseen and Qirem 2018) is consistent with the study of consumers' continuity of the use of the payment system and their acceptance of costs and experience. Accordingly, our research on the intention and acceptance of digital payment systems is based on the UTAUT model. A study conducted by UTAUT identified four key constructs, i.e., performance and effort expectancies, social influence, and facilitating conditions, that influence behavior related to the behavioral intention to use technology (Venkatesh et al. 2012).

The present study is primarily designed to identify the factors that facilitate users' acceptance of new technologies. A contribution of this study is the analysis of variables examined in traditional models UTAUT to determine the direct and indirect effects of determinants related to the adoption of e-payment systems. Additionally, this research employs a set of variables widely used in the scientific literature in most countries within UTAUT but not in Yemen. Furthermore, this study examines the most popular and widely accepted e-payment methods perceived by smartphones, tablets, computers, and desktop users. Overall, a clear research purpose is essential for predicting customer behavior towards e-payment systems currently available in different markets and assessing possible future intentions associated with this study area.

The rest of this paper is organized as follows: Section 2 reviews the literature and develops hypotheses. Section 3 describes the methodology of the study. Section 4 presents the data analysis and results. The discussion is presented in Section 5. The final section, Section 6, outlines the research conclusions, contributions, limitations, and recommendations for future research.

## 2. Literature Review and Hypotheses Development

Since the Unified Theory of Acceptance and Use of Technology (UTAUT) was developed by (Venkatesh et al. 2003), it has been composed of four constructs. This model consists of the same four constructs: performance expectancy (PE), effort expectancy (EE), social influence (SI), and facilitating conditions (FCS). These four factors are directly associated with user acceptance of an e-payment system and usage behavior. They are concerned with the users' perception concerning the usefulness of an e-payment system (Lutfi 2022). The UTAUT model has been tested and used in many previous studies (Yaseen and Qirem 2018; Abushanab and Pearson 2007; Al-Somali et al. 2009; Riffai et al. 2012; Nasri and Charfeddine 2012; Lutfi 2022). There is no doubt that the UTAUT model is highly effective

in examining how technology is adopted in literature, particularly in determining the factors that influence the intention to use and actual use of technology (Venkatesh et al. 2003; Venkatesh et al. 2012). Hence, this study adopts the UTAUT model for understanding the adoption and usage of digital payment systems. (Alsyouf and Ishak 2018; Lutfi 2022) illustrated how the UTAUT captures and enfolds the concept of technology acceptance into a single view, combining theories and models related to technology acceptance.

The purpose of this study is to investigate the intention of customers in Yemen to use e-payment using the UTAUT by testing hypotheses concerning the effects of the UTAUT factors on the behavioral intention to use the e-payments. Moreover, this hypothetical model (see Figure 1), derived from a literature review, explains the relationship between the four factors and consumers' behavioral intention to accept and use e-payments (Venkatesh et al. 2003). According to the structural model, the relationship is moderated by gender and age in the second stage.

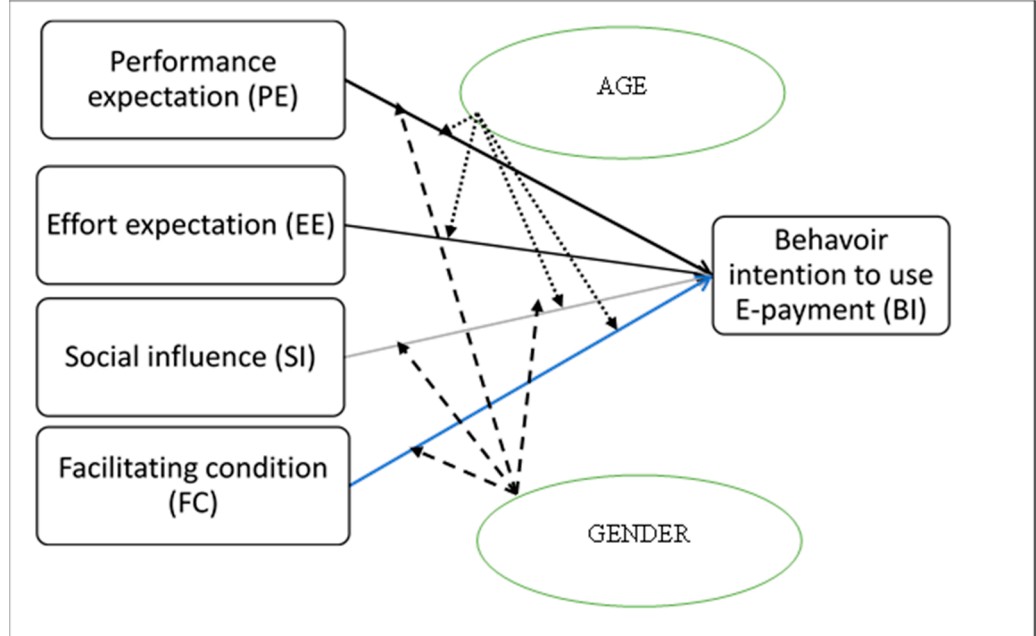

**Figure 1.** The proposed theoretical model.

*2.1. Factors Affecting the Intention to Use E-Payment*

The UTAUT model is useful in studying the factors influencing technology adoption (Venkatesh et al. 2003, 2011). In the IT field, the UTAUT model combines theoretical and experimental findings concerning user acceptance relationships (Lutfi 2022; Al-Okaily et al. 2020). This study considers the intention to use rather than continuance intention as a dependent factor since the sample frame consists of Yemeni consumers who use electronic payment systems. Currently, the use of e-payment systems in Yemen just takes place through banking applications, which are still in their infancy, and purchases through e-payment systems are pretty limited.

2.1.1. Performance Expectancy

According to (Venkatesh et al. 2003), performance expectancy is the perception that a particular technology will boost the efficiency and effectiveness of a task's performance. There are several components of PE, such as usefulness (Davis et al. 1989; Davis 1989), intrinsic motivation (Davis et al. 1992), job fit (Thompson et al. 1991), and outcome expectations (Compeau and Higgins 1995; Compeau et al. 1999). According to (Davis et al. 1992), perceived usefulness refers to an individual's expectation that technology use will enhance their performance at work. This study uses the term PE to describe consumers' perceptions that using e-payments will improve their efficiency and effectiveness in accomplishing their

work tasks and the quality of services provided. Thus, the authors have considered the PE of new information technology influencing the intention to use e-payments (Koenig-Lewis et al. 2015; Gupta et al. 2020; Ariffin and Lim 2020). According to most research findings, performance expectancy positively and significantly impacts consumers' intentions (Ariffin and Lim 2020; Venkatesh et al. 2003; Sivathanu 2019; Schierz et al. 2010; Martins et al. 2014; Oliveira et al. 2014). The following hypothesis is proposed:

**H1:** *Performance expectancy significantly and positively affects the BI to use of e-payments.*

### 2.1.2. Effort Expectancy

Effort expectancy refers to the ease of using a system (Venkatesh et al. 2003, 2011). It is most commonly explained through three fundamental concepts: complexity, ease of use, and perceived ease of use. (Venkatesh and Davis 2000) identified general factors influencing the perceived usefulness, and (Venkatesh 2000) identified factors influencing the perceived ease of use. According to (Moore and Benbasat 1991), ease of use refers to an individual's perception that the use of a system is free of mental and physical effort, while complexity is the perception of users that using technology is difficult to understand (Thompson et al. 1991). The effort expectancy in this study refers to the perception of ease of use of e-payments. The literature review indicates that effort expectancy significantly affects the intention to use an e-payments system (Venkatesh et al. 2003; Martins et al. 2014; Oliveira et al. 2014; Sivathanu 2019; Zhou et al. 2010; Koksal 2016; Yaseen and Qirem 2018) and continuance intention of customers to use e-payments system (Venkatesh et al. 2011; Almaiah et al. 2021; Lutfi 2022). Therefore, the following hypothesis is proposed:

**H2:** *Effort expectancy significantly and positively affects the BI to use e-payments.*

### 2.1.3. Social Influence

The concept of social influence is defined by (Venkatesh et al. 2003) as the perception by others that an individual needs to use a new system. It consists of an image (Moore and Benbasat 1991), subjective norms (Ajzen 1991; Davis 1989; Taylor and Todd 1995a, 1995b), and social factors (Thompson et al. 1991). Social influences in the form of subjective norms play a significant role in models of the technology acceptance model and its subsequent adaptations (Venkatesh and Bala 2008). This factor is how people's perceptions of what matters to them influence their decision to adopt a certain system or perform a specific action (Venkatesh and Bala 2008). If a particular payment system can attract the attention of users considered as references by influencers, it is likely to be adopted by others on time in the future. In general, this result significantly impacts the success of e-payment systems. According to the research model, social influence means the extent to which customers believe they should accept and use e-payments. Previous studies found that SI has a significant impact on BI (Koksal 2016; Oliveira et al. 2014; Sivathanu 2019; Malaquias and Hwang 2019; Al-Okaily et al. 2020; Alsyouf and Ishak 2018; Ariffin et al. 2021; Martins et al. 2014). As a result, we develop the following hypothesis:

**H3:** *Social influence significantly and positively affect the BI to use e-payments.*

### 2.1.4. Facilitating Conditions

According to (Venkatesh et al. 2003), facilitating conditions describe how much an individual believes that organizational and technical infrastructure exists to facilitate the use of a system. This definition encompasses three different constructs: perceived behavioral control (Taylor and Todd 1995a, 1995b; Ajzen 1991), facilitating conditions, and compatibility (Thompson et al. 1991). According to the current study, facilitating conditions refer to customer perceptions that specific factors in the infrastructure of the e-payment system either prevent or promote the acceptance and use of e-payment. Based on the findings of (Zhou et al. 2010; Sivathanu 2019), FC is positively associated with using mobile

payment. In our study, we hypothesized that it might be positively associated with BI to the use of e-payment. Therefore, we propose the following hypothesis:

**H4:** *Facilitating conditions significantly and positively influence the BI to use e-payments.*

*2.2. Impact of the Factors on Intentions to Use E-Payment Moderating by Gender and Age*

In our study, we adopted gender and age from (Venkatesh et al. 2003) to moderate the relationship between PE, EE, SI, and FCs and behavioral intention. Based on (Venkatesh et al. 2003), gender, age, experience, and voluntariness of use moderate the relationships between behavioral intention and other factors. Then we hypothesize the following:

**H5:** *The impact of performance expectancy on behavioral intention will be moderated by gender and age.*

**H6:** *The impact of effort expectancy on behavioral intention will be moderated by gender and age*

**H7:** *The impact of social influence on behavioral intention will be moderated by gender and age.*

**H8:** *The impact of facilitating conditions on behavioral intention will be moderated by gender and age.*

## 3. Research Design and Methodology

*3.1. Sample*

This paper aimed to find the general intention of Yemeni customers concerning digital payments. The study sample included 494 Yemeni, focusing on the potential of consumers using e-payments as a new payment system in Yemen. The population in this study is identified as smartphone users from all Yemeni districts. In consideration of the feasibility and cost-effectiveness of the research, a convenient online survey tool (Google Forms) was used to collect the data. The respondents were surveyed via a self-administered survey from 18 June 2022 to 5 August 2022. The final sample after removing the missing data was 486, as in Table 1:

**Table 1.** Sample description.

| Description | Responses |
|---|---|
| Total received responses | 493 |
| Missing data | −7 |
| **Final sample** | **486** |

Source: (The authors).

This study used an e-survey for the convenience of collecting data. The survey link was circulated over social media platforms. The questionnaire was structured in two sections. Section A was dedicated to collecting demographic information, and Section B was dedicated to measuring all variables. We measured all items using Likert scales (1 = Strongly Disagree, and 5 = Strongly Agree).

*3.2. Measure*

UTAUT was considered with four constructs as shoeing in Table 2: Performance Expectation (PE: five items) adapted from (Martins et al. 2014; Zhou 2012; Im et al. 2011; Zhou et al. 2010; Abrahão et al. 2016; Venkatesh et al. 2003), effort expectation (EE: four items) adapted from (Davis et al. 1989; Venkatesh et al. 2003; Venkatesh et al. 2012; Zhou et al. 2010; Abrahão et al. 2016; Zhou 2012; Martins et al. 2014), social influence (SI: four items) adapted from (Venkatesh et al. 2003; Gu et al. 2009; Zhou et al. 2010; Im et al. 2011; Zhou 2012; Martins et al. 2014), and facilitating condition (FCS: three items) adapted from (Venkatesh et al. 2003), to discuss the behavioural intention to use the e-payment system (BI: three items) adapted from (Lu et al. 2011; Yang et al. 2012; Lin 2011; Schierz et al. 2010; Gu et al. 2009; Abrahão et al. 2016; Zhou et al. 2010; Im et al. 2011; Yaseen and Qirem 2018).

**Table 2.** The research constructs.

| Construct | Code | Items |
|---|---|---|
| Performance expectation (PE) | PE1 | I believe e-payments would be a useful service in my day-to-day activities. |
| | PE2 | Using e-payments would make me perform my financial transactions more quickly. |
| | PE3 | Using e-payments would save time so that I can do other activities in my day-to-day. |
| | PE4 | E-payment would bring me greater convenience. |
| | PE5 | The use of an e-payment system will enhance the completion of many tasks per day. |
| Effort expectation (EE) | EE1 | My interaction via e-payment would be clear and easy to understand. |
| | EE2 | It would be easy for me to develop the skills needed to use e-payment. |
| | EE3 | I believe that it is easy to use e-payment. |
| | EE4 | Learning to use e-payment would be easy for me. |
| Social influence (SI) | SI1 | People who influence my behavior would think I should use e-payment (when available). |
| | SI2 | People who are important to me would think that I should use e-payment (when available). |
| | SI3 | People who are important to me could assist me in the use of e-payment (when available). |
| | SI4 | People whose opinions I appreciate prefer to use the e-payment system. |
| Facilitating conditions | FC1 | The necessary resources are available for the use of the e-payment system (fast internet, modern devices). |
| | FC2 | I have the necessary knowledge to use the e-payment system. |
| | FC3 | A specific person (or group) is available to assist with the difficulties of the e-payment system. |
| Behavioral intention (BI) | BI1 | I intend to use e-payment services in the future. |
| | BI2 | I will always try to use the e-payment system in my daily life. |
| | BI3 | I plan to use the e-payment system in the future. |

*3.3. Procedures and Analysis*

There were 24 items attached through a supplementary file at (https://drive.google.com/file/d/1XKyMC_0wi68fWgcvL0GfYQwBqd7r5yzD/view?usp=sharing, accessed on 1 August 2022). After the survey's first draft had been completed, the co-authors shared this for evaluation against the study's objectives; they completed a trial version of the survey and provided feedback on its readability and usability. Linguistic and technical modifications were applied until an agreement was reached on the final version, although the survey content did not change. Checks were performed on both the Arabic and English versions, although only the Arabic version was used for data collection. Statistically, the data was analyzed by Jamovi software (Revelle 2019; Core Development Team 2021; Jamovi Project 2021). However, the structural model analysis was employed using smartPLS4 and Microsoft Excel. Several regression analyses were conducted to examine the relationship between predictors and dependent variables. Before conducting the multivariate regression analysis, several assumptions were examined concerning normality and multicollinearity. Furthermore, we conducted some analyses of sample characteristics based on ratios and frequencies.

*3.4. Model Specification*

Based on the regression analysis method, Equation (1) was used to analyze the impact of UTAUT factors (performance expectancy [PS], effort expectancy [EE], and facilitating conditions [FCs]) on behavioral intention (BI):

$$\text{BI} = f\left(\sum \text{UTAUT Factors}\right) + \varepsilon \tag{1}$$

Further analysis is provided in Equations (2) and (3). We examined the impact of UTAUT factors (performance expectations [PS], effort expectations [EE], and facilitating conditions [FCs]) on behavioral intentions (BI)) moderated by gender and age.

$$\text{BI} = f\left(\sum \text{UTAUT Factors*GENDER}\right) + \varepsilon \tag{2}$$

$$\text{BI} = f\left(\sum \text{UTAUT Factors*AGE}\right) + \varepsilon \tag{3}$$

## 4. Results

*4.1. Demographic Characteristics of Participants*

As shown in Table 3, among the sample, the majority were males (85.4%), whereas females were 14.6%, which is not surprising since the majority of the Yemeni banks' clients are men (Al-Ajam and Nor 2015). In Yemeni society, it is well known that men play an essential role in the social and economic life of the country.

**Table 3.** Participants' characteristics.

| Socio-Academic Characteristics | Total (%) |
|:---:|:---:|
| **Age range** | |
| <18 | 1 (0.2) |
| 18–24 | 84 (17.3) |
| 25–35 | 153 (31.5) |
| 36–45 | 174 (35.8) |
| 46–55 | 46 (9.5) |
| ≥56 | 28 (5.8) |
| Gender | |
| **Male** | 353 (72.6) |
| Female | 133 (27.4) |

*4.2. Descriptive and Correlation Analysis*

Table 4 summarizes the sample statistics. To analyze our descriptive statistics, we tested the normality of the sample by using the Shapiro–Wilk test, which indicated that all variables had a p-value of 0.001. PE, EE, SI, and FCs had respective means of 4.31, 4.26, 4.03, and 3.51 for each factor. Additionally, the mean behavioral intention was 4.32, which is very high. However, we note that FCs had the lowest mean, and this is because one of the items of FCs (FCs1) impaired the other factors, as well as the behavioral intention to use electronic payments. Respondents indicated that they do not possess the appropriate infrastructure to accept electronic payments, including the internet speed and the payment process quality, as shown in Table 5. An analysis of the matrix correlations of the variables is presented in Table 4. Except for the FCs factor, all other variables have a significant positive correlation.

**Table 4.** Descriptive statistics.

| Variables | N | Mean | SD | Shapiro-Wilk (Normality) | |
|---|---|---|---|---|---|
| | | | | W | *p* |
| PE | 486 | 4.31 | 0.631 | 0.767 | <0.001 |
| EE | 486 | 4.26 | 0.67 | 0.763 | <0.001 |
| SI | 486 | 4.03 | 0.848 | 0.862 | <0.001 |
| FCs | 486 | 3.51 | 0.651 | 0.907 | <0.001 |
| BI | 486 | 4.32 | 0.666 | 0.751 | <0.001 |

Note. PE is the performance expectancy. EE is the effort expectancy. SI is the social influence. FCs are the facilitating conditions.

**Table 5.** Empirical correlation matrix.

| Variable | BI | PE | EE | SI | FCS | FCS1 | FCS2 | FCS3 |
|---|---|---|---|---|---|---|---|---|
| BI | — | | | | | | | |
| PE | 0.631 *** | — | | | | | | |
| EE | 0.638 *** | 0.616 *** | — | | | | | |
| SI | 0.568 *** | 0.558 *** | 0.716 *** | — | | | | |
| FCs | 0.240 *** | 0.291 *** | 0.251 *** | 0.180 *** | — | | | |
| FCs1 | −0.053 | −0.167 *** | −0.140 ** | −0.209 *** | 0.496 *** | — | | |
| FCs2 | 0.492 *** | 0.484 *** | 0.414 *** | 0.320 *** | 0.555 *** | −0.128 ** | — | |
| FCs3 | 0.268 *** | 0.473 *** | 0.490 *** | 0.494 *** | 0.627 *** | −0.180 *** | 0.478 *** | — |

Note. PE is the performance expectancy. EE is the effort expectancy. SI is the social influence. FCs are the facilitating conditions. ** $p < 0.01$, *** $p < 0.001$.

### 4.3. Measurement Model

Analyses of confirmatory factors confirmed the initial factor structure. It was planned that indicators that failed to meet the criteria would be eliminated successively in this study. Data analysis was based on the accepted criteria of factor loading greater than 0.5 and a well-explained factor structure. Based on Table 6, no items were required to be eliminated from the present study.

**Table 6.** Outer loadings matrix.

| Construct | 1 | 2 | 2 | 4 | 5 |
|---|---|---|---|---|---|
| BI1 | 0.904 | | | | |
| BI2 | 0.738 | | | | |
| BI3 | 0.777 | | | | |
| EE1 | | 0.818 | | | |
| EE2 | | 0.797 | | | |
| EE3 | | 0.792 | | | |
| EE4 | | 0.748 | | | |
| FC1 | | | 0.571 | | |
| FC2 | | | 0.894 | | |
| FC3 | | | 0.608 | | |
| PE1 | | | | 0.822 | |
| PE2 | | | | 0.813 | |
| PE3 | | | | 0.748 | |
| PE4 | | | | 0.818 | |
| PE5 | | | | 0.758 | |
| SI1 | | | | | 0.82 |
| SI2 | | | | | 0.787 |
| SI3 | | | | | 0.799 |
| SI4 | | | | | 0.887 |

Note. PE is the performance expectancy. EE is the effort expectancy. SI is the social influence. FCs are the facilitating conditions.

Additionally, this study examines the validity, reliability of the outer model, and discriminant validity (Fornell–Larcker criterion) following (Hair et al. 2019). Composite reliability (CR), and convergent validity (the average variance derived [AVE]). Table 7 presents the results of the outer model. Moreover, it illustrated that the CR and AVE are all greater than 0.7 and 0.5, except FC, the CR and AVE are 0.675 and 0.424. Thus, we can accept 0.424. Because according to (Fornell and Larcker 1981), if the AVE is less than 0.5 and the composite reliability is greater than 0.6, the convergent validity of the construct is still acceptable.

**Table 7.** Reliability, convergent, and discriminant validities.

| | Construct Reliability and Validity | | | Discriminant Validity (Fornell–Larcker Criterion) | | | | |
|---|---|---|---|---|---|---|---|---|
| Variable | $\alpha$ | CR | AVE | BI | PE | EE | SI | FCs |
| BI | 0.850 | 0.850 | 0.655 | 0.810 | | | | |
| PE | 0.894 | 0.894 | 0.628 | 0.923 | 0.792 | | | |
| EE | 0.868 | 0.868 | 0.623 | 0.931 | 0.911 | 0.789 | | |
| SI | 0.895 | 0.894 | 0.679 | 0.780 | 0.773 | 0.875 | 0.824 | |
| FCs | 0.739 | 0.675 | 0.424 | 0.754 | 0.784 | 0.820 | 0.716 | 0.705 |

Note. $\alpha$: Cronbach's a; CR: composite reliability; AVE: average variance extracted. PE is the performance expectancy. EE is the effort expectancy. SI is the social influence. FCs are the facilitating conditions.

### 4.4. Structural Model

Our next step is to test the structural model after completing the evaluation of the measurement model. To evaluate the degree of collinearity between the indicators, the variance inflation factor (VIF) is commonly used. Initially, the VIF was measured. (Hair et al. 2019) recommends that the VIF be less than or close to 3. According to our findings, the VIF was less than 3 for all constructs. Based on (Hair et al. 2019), $R^2$ values of 0.75, 0.50, and 0.25 are considered substantial, moderate, and weak, respectively. According to the $R^2$ values for BI (0.757), this is considered a substantial value for measuring variance.

### 4.5. Hypothesis Test

#### 4.5.1. The Impact of the Factors of UTAUT on BI

Table 8 presents the regression results with equation 1 in columns 1–4. Among all UTAUT factors, the estimated coefficients are positive and significant at $p < 0.001$. According to our expectations, there is a higher acceptance of using e-payment. Accordingly, these findings supported our hypothesis H1–H4. The PE ($\beta = 0.848$, $p < 0.001$), had a direct positive and significant effect on customer's intention to use e-payments. The EE ($\beta = 0.794$, $p < 0.001$) positively and significantly affected customers' behavioral intention to use e-payment. The SI ($\beta = 0.529$, $p < 0.001$) was also found to affect the behavioral intention significantly to use e-payment. Although FCs were also found to be statistically significant ($\beta = 0.527$, $p < 0.001$) to have a direct positive and significant effect on customers' behavioral intention to use electronic payment. After additional testing of FCs items in column 5, FCs1 had a direct negative and significant effect on customers' behavioral intention to use electronic payment.

#### 4.5.2. Further Analysis—Testing the Moderating Effect

According to Table 9, the moderator effect of age and gender plays a significant role in identifying the significance of age and gender in the moderated multiple regression analysis results. Moreover, the results indicated that age was not a moderating variable between PE ($\beta = 0.007$ is not significant), SI ($\beta = 0.010$ is not significant), and behavior intention. However, the two constructs EE ($\beta = 0.010$, $p < 0.05$) and FCs ($\beta = 0.019$, $p < 0.05$) were significantly moderated by age. In addition, the findings indicate that gender is a moderating variable between PE ($\beta = 0.042$, $p < 0.001$), EE ($\beta = 0.043$ $p < 0.001$), SI ($\beta = 0.044$ $p < 0.001$), FCs ($\beta = 0.071$ $p < 0.001$) and behaviour intention. The previous results support some of the findings of the UTAUT model regarding the BI.

**Table 8.** Factors influencing behavioral intention.

| Model | (1) | (2) | (3) | (4) | (5) |
|---|---|---|---|---|---|
| **Variable** | **BI** | **BI** | **BI** | **BI** | **BI** |
| PE | 0.848 *** | | | | |
| | (0.029) | | | | |
| EE | | 0.794 *** | | | |
| | | (0.027) | | | |
| SI | | | 0.529 *** | | |
| | | | (0.026) | | |
| FCs | | | | 0.527 *** | |
| | | | | (0.040) | |
| FCs1 | | | | | −0.176 *** |
| | | | | | (0.025) |
| FCs2 | | | | | 0.403 *** |
| | | | | | (0.040) |
| FCs3 | | | | | 0.058 * |
| | | | | | (0.028) |
| Constant | 0.665 *** | 0.942 *** | 2.193 *** | 2.473 *** | 2.318 *** |
| | (0.1245) | (0.117) | (0.109) | (0.123) | (0.138) |
| Observations | 486 | 486 | 486 | 486 | 486 |
| R | 0.804 | 0.799 | 0.673 | 0.515 | 0.577 |
| $R^2$ | 0.646 | 0.6639 | 0.453 | 0.265 | 0.333 |

Note. PE is the performance expectancy. EE is the effort expectancy. SI is the social influence. FCs are the facilitating conditions. * $p < 0.05$, *** $p < 0.001$.

**Table 9.** The moderating effect on behavioral intention.

| Model | (1) | (2) | (3) | (4) |
|---|---|---|---|---|
| **Variables** | **BI** | **BI** | **BI** | **BI** |
| PE | 0.755 *** | | | |
| | (0.038) | | | |
| PE*Age | 0.007 | | | |
| | (0.004) | | | |
| PE*Gender | 0.042 *** | | | |
| | (0.010) | | | |
| EE | | 0.691 *** | | |
| | | (0.0370) | | |
| EE*Age | | 0.010 * | | |
| | | (0.004) | | |
| EE*Gender | | 0.043 *** | | |
| | | (0.010) | | |
| SI | | | 0.424 *** | |
| | | | (0.042) | |
| SI*Age | | | 0.010 | |
| | | | (0.005) | |
| SI*Gender | | | 0.044 *** | |
| | | | (0.013) | |
| FCs | | | | 0.359 *** |
| | | | | (0.056) |
| FCs*Age | | | | 0.019 * |
| | | | | (0.007) |
| FCs*Gender | | | | 0.071 *** |
| | | | | (0.017) |
| Constant | 0.724 *** | 0.998 *** | 2.245 *** | 2.509 *** |
| | (0.123) | (0.116) | (0.109) | (0.141) |
| Observations | 486 | 486 | 486 | 486 |
| R | 0.812 | 0.808 | 0.683 | 0.541 |
| $R^2$ | 0.659 | 0.652 | 0.467 | 0.293 |

Note. PE is the performance expectancy. EE is the effort expectancy. SI is the social influence. FCs are the facilitating conditions. * $p < 0.05$, *** $p < 0.001$.

## 5. Discussion

According to the structural model findings, $R^2 = 0.757$ explained the variance of behavioral intentions across all constructs using PLS-SEM by calculating the path coefficients and their significance (Figure 2). Statistical analyses indicated that the intention to use e-payments was significantly and positively influenced by performance, expectancies, and social influences, which are consistent with previous studies such as (Venkatesh et al. 2003; Martins et al. 2014; Oliveira et al. 2014; Sivathanu 2019; Zhou et al. 2010; Koksal 2016; Abrahão et al. 2016; Koenig-Lewis et al. 2015). On the contrary, facilitation conditions were significantly and negatively correlated with behavioral intention, which was inconsistent with (Zhou et al. 2010; Sivathanu 2019; Ting et al. 2016). This might be attributed to consumers' perceptions of the infrastructure of internet services, which does not encourage consumers to accept electronic payments in Yemen. (Yaseen and Qirem 2018; Ariffin and Lim 2020) indicated that PE was not a significant predictor, which was inconsistent with our findings.

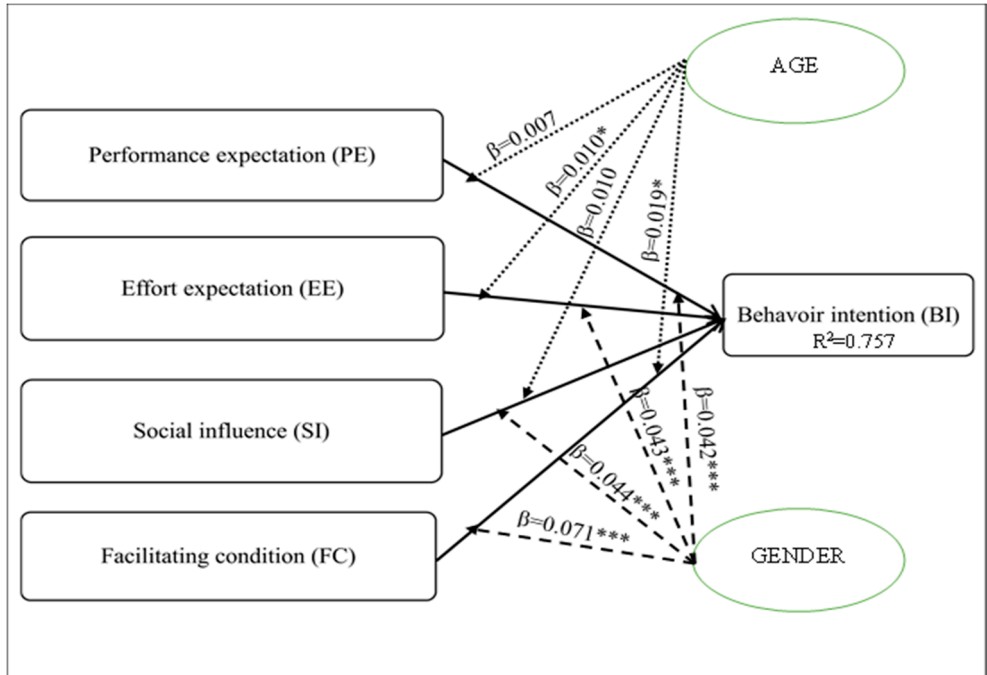

**Figure 2.** The structural model results. Note: * $p < 0.05$, *** $p < 0.001$.

Contrary to expectations, the relationship between performance expectancy, social influence, and behavioral intention to use e-payment systems was not moderated by age; therefore, the related hypothesis is not supported. Our findings were inconsistent with (Koenig-Lewis et al. 2015; Wu and Wang 2005; Koufaris 2002); they have previously found insignificant relationships between ease to use (represents a construct of TAM theory) and behavioral intention, which has been shown in our results as effort expectancy based on the UTAUT model. Moreover, we found evidence that customers have neither acceptance nor intention to use electronic payment as long as the infrastructure is inadequate. They think the Internet is not fast and its quality is poor, which does not help them access e-payments. Our results show that consumers will use less traditional and digital payment methods in the coming years, consistent with (Qin et al. 2017; Liébana-Cabanillas et al. 2014; Luna et al. 2019).

According to the study, gender significantly influenced the relationship between factors and intention to accept electronic payments. However, the results indicated that age did not affect the relationship between performance expectancy, behavior influence, and the intention to use e-payments, which is in line with (Riskinanto et al. 2017), who indicated

that most of the constructs' associations are not affected by age. In addition, our findings revealed that the factor of facilitating conditions is the most crucial factor and the most important motivation for electronic payment usage. The study found that the performance expectation factor had a higher significance and a higher positive significance than the other factors, which is in line with (Abrahão et al. 2016). Moreover, the findings suggested that users will likely use this feature if it is available. This study developed a model for predicting the intention to accept electronic payments and tested it in a country that has not given enough attention to this topic. The UTAUT model was applied and tested to the electronic payment system in Yemen. Finally, the study clarified that the infrastructure in Yemen significantly impacts the intention to use electronic payment.

## 6. Conclusions

E-payments have recently gained popularity as a result of the increasing use of smartphones and their applications all over the world. There are few studies on this technology in Yemen, such as (Al-Ajam and Nor 2015). This study highlights the importance of examining the primary factors that affect Yemeni users' decisions to adopt and use an e-payment system. Based on the UTAUT model, the study examined behavioral intention as an independent variable and gender and age as moderating variables (Venkatesh et al. 2003). Based on the findings, it is possible to draw a few conclusions. The first finding confirms some of the UTAUT model's predictions, although all predictors were significant. According to the research findings, performance and effort expectancies and social influence were significant and positive predictors of the intention to accept e-payments. However, facilitation conditions are significant and negative because they indicate that consumers are dissatisfied with the infrastructure of Yemen's Internet. The main statistical results of the study confirmed the predictive validity of the research model. Furthermore, they explain ($R^2$: 0.757) the variance in the intention to use e-payment systems.

The findings have several practical implications for researchers, practitioners, and policymakers. Because the abolishment of traditional payment systems is a rare phenomenon that occurs in any economy, the findings of this study provide economists with valuable insights for designing a transition matrix for a smooth transition from cash-based payment systems to cashless digital payment systems. It is essential for the long-term sustainable adoption of digital payment systems that they consider consumer desires, meet their needs, and provide the infrastructure needed for their use. The government has a crucial role in increasing awareness, promoting digital literacy among consumers, and facilitating adoption by providing supportive infrastructure. By doing so, the transition from traditional to digital payment systems will be more accessible. Furthermore, this study provides valuable insights to system developers and business decision-makers regarding the services they provide to consumers, as well as to the telecommunications authority and other companies operating in the telecommunications industry in Yemen. Studying the intention of using electronic payment in general, we focused on users and target consumers who can access the Internet through a personal computer, laptop, tablet, or smartphone. We concentrated on the UTAUT theory only because the study was applied in Yemen, and the e-payment service is still not widely accepted without the infrastructure that contributes to attracting and protecting consumers (Qin et al. 2017).

Furthermore, the results of this study offer several recommendations and suggestions for researchers and practitioners in the field of e-payment technology development to facilitate a greater acceptance of mobile payments. The study recommends that academics and researchers perform complementary studies in this field and continue to explore the determinants of the intention to use e-payment, as there is a lack of research in this area. Due to the present significance of electronic payment systems and the need to keep up with technological developments in other countries, researchers should consider promoting research on using electronic payment systems in Yemen. Moreover, telecommunications companies should promote the use of the Internet, encourage citizens to download applications for electronic payment, and provide the necessary protection throughout the

purchasing process. Furthermore, since the study results explained 75% of the variance in the intention of use among the participants, further research should be conducted to incorporate variables from other theories, including ease to use, perceived risk, security, etc. Providing the basic requirements to use the digital payment system is necessary, and the monopoly of competent authorities on the Internet has contributed to the delay in using the digital payment system over the phone and the lack of interest in the current payment systems through using banking services.

**Supplementary Materials:** The following supporting information can be downloaded at: https://drive.google.com/file/d/1XKyMC_0wi68fWgcvL0GfYQwBqd7r5yzD/view (accessed on 1 August 2022).

**Author Contributions:** Conceptualization, F.A. and M.O.A.-S.; Data curation, F.A.; Formal analysis, F.A.; Funding acquisition, F.A. and M.O.A.-S.; Investigation, F.A. and M.O.A.-S.; Methodology, F.A. and M.O.A.-S.; Project administration, F.A.; Software, F.A.; Supervision, F.A.; Validation, F.A. and M.O.A.-S.; Visualization, F.A. and M.O.A.-S.; Writing—original draft, F.A. and M.O.A.-S.; Writing—review and editing, F.A. and M.O.A.-S. All authors have read and agreed to the published version of the manuscript.

**Funding:** This research received no external funding.

**Institutional Review Board Statement:** All subjects gave their informed consent for inclusion before participating in the study. The study was conducted in accordance with the Declaration of Helsinki, and the Ethics Committee approved the protocol of the Department of Accounting, Faculty of Business, Philadelphia University, Jarash Road, 20 KM, Amman Jordan on 31 May 2022.

**Informed Consent Statement:** Informed consent was obtained from all subjects involved in the study.

**Data Availability Statement:** The data presented in this study is available on request from the first author. The data are not publicly available due to ethical restrictions.

**Acknowledgments:** The publication of this research has been supported by the Deanship of Scientific Research and Graduate Studies at Philadelphia University—Jordan.

**Conflicts of Interest:** The authors declare no conflict of interest.

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
