# Peer review of "Intention to Use E-Payments from the Perspective of the Unified Theory of Acceptance and Use of Technology (UTAUT): Evidence from Yemen"

_economies, doi:10.3390/economies10100259_

Round 1

Reviewer 1 Report

The paper addresses an important and topical issue attempting to investigate the determinants of e-payments adoption in an emerging economy of Yemen. Not withstanding the overall scientific soundness of the paper, however, the Authors might consider the following suggestions to improve its clarity and consistency:

·      Although the majority of items used to measure the key variables of the model are briefly mentioned during the hypothesis development, it would be highly advisable to include their detailed explanation (preferably in the form of an additional table in section 3.2). In the current version of the paper it is not clear what the particular 'items' stand for and how they were measured. In particular, it seems that the items used to proxy Behavioral Intentions (BI) have not been discussed at all.

Providing more details about individual constructs used to capture the latent factors may also be crucial for correct interpretation of the results of the estimation and their subsequent discussion.

The above issues seem particularly important in the light of several unexpected (and at the same time inconsistent with prior studies) results reported by the Authors - see the discussion section.

·      at the beginning of section 4.1 it is not necessary to repeat the exact content of Table 2 in the main text,

·      The style of references is inconsistent (see e.g. line 34),

·      Although the language of the paper is generally fine and understandable some further proof reading seems advisable with respect to wording, grammar, and style, see e.g.:

-       in the title of the paper using a plural form ‘payments’ or adding the word ‘systems’ seem more suitable,

-       in the formulation of the hypothesis H1 (line 115) the word ‘to’ seems to be missing (it should be: ‘… BI to use…’),

-       line 125 – ‘… continuance the intention…’;

-       lines 146-147 – the sentence contains several repetitions and the word ‘definitionis’,

-       Table 1, p. 5 ‘missed’ instead of ‘missing’,

-       line 228 – ‘are less mean’,

-       line 242 – ‘confirmed by confirmatory…’.

Author Response

I would like to thank you for your time and effort in helping me improve this paper's quality. Thank you for your comments, and I hope my revisions and modifications will meet your expectations.

Kindly find attached the response to your comments.

Reviewer 2 Report

This study examines the intention to use E-payment system in Yemen. I think it's an interesting work in this financial technology era. But, there are some issues that can be improved in my opinion. The questions are as below:

1. The examination of intention to use in E-payment or mobile payment system field is more evidence. So, I suggest authors can improve the important things and reasons in the case study of Yemen.

2. The theoretical background of this work was applied the "UTAUT". However, the "UTAUT2" was proposed by Venkatesh et al., in 2012. I suggest authors should make more explanations on this issue. 

3. In the research framework, authors combine the "behavior and intention to use" concepts with BI. However, the UTAUT2 separated the concept of BI in two things which include the "behavioral intention" and "use behavior". Hence, my opinion in this field should be making more explanations the reasons.

4. In conclusion, I suggest authors can improve their contributions on academically and commercially.

Author Response

(The authors gave the same response as above.)

Round 2

Reviewer 2 Report

Authors are answered the all of my questions. 

I have no any problems in this work.